# Acute Toxicity of Carbon Nanotubes, Carbon Nanodots, and Cell-Penetrating Peptides to Freshwater Cyanobacteria

**DOI:** 10.3390/toxins17040172

**Published:** 2025-04-01

**Authors:** Anna K. Antrim, Ilana N. Tseytlin, Emily G. Cooley, P. U. Ashvin Iresh Fernando, Natalie D. Barker, Erik M. Alberts, Johanna Jernberg, Gilbert K. Kosgei, Ping Gong

**Affiliations:** 1Environmental Laboratory, U.S. Army Engineer Research and Development Center, 3909 Halls Ferry Road, Vicksburg, MS 39180, USA; anna.k.antrim@usace.army.mil (A.K.A.); natalie.d.barker@usace.army.mil (N.D.B.); gilbert.k.kosgei@usace.army.mil (G.K.K.); 2Oak Ridge Institute for Science and Education, 1299 Bethel Valley Rd, Oak Ridge, TN 37830, USA; ilanatse@pitt.edu (I.N.T.);; 3Bennett Aerospace, Inc., 1 Glenwood Ave, Raleigh, NC 27603, USA; 4SIMETRI, Inc., 937 S Semoran Blvd, Suite 100, Winter Park, FL 32792, USA

**Keywords:** harmful cyanobacterial bloom (HCB), non-metallic nanoparticles (NMNPs), carbon nanodots (CNDs), single-walled carbon nanotubes (SWCNTs), cell-penetrating peptides (CPPs), acute toxicity, freshwater cyanobacteria, mitigation, cell density, chlorophyll-a, phycocyanin, photosynthesis efficiency

## Abstract

Synthetic non-metallic nanoparticles (NMNPs) such as carbon nanotubes (CNTs), carbon nanodots (CNDs), and cell-penetrating peptides (CPPs) have been explored to treat harmful algal blooms. However, their strain-specific algicidal activities have been rarely investigated. Here we determined their acute toxicity to nine freshwater cyanobacterial strains belonging to seven genera, including *Microcystis aeruginosa* UTEX 2386, *M. aeruginosa* UTEX 2385, *M. aeruginosa* LE3, *Anabaena cylindrica* PCC 7122, *Aphanizomenon* sp. NZ, *Planktothrix agardhii* SB 1810, *Synechocystis* sp. PCC 6803, *Lyngbya* sp. CCAP 1446/10, and *Microcoleus autumnale* CAWBG635 ATX. We prepared in-house three batches of CNDs using glucose (CND-G) or chloroform and methanol (CND-C/M) as the substrate and one batch of single-walled CNTs (SWCNTs). We also ordered a commercially synthesized CPP called γ-Zein-CADY. The axenic laboratory culture of each cyanobacterial strain was exposed to an NMNP at two dosage levels (high and low, with high = 2 × low) for 48 h, followed by measurement of five endpoints. The endpoints were optical density (OD) at 680 nm (OD_680_) for chlorophyll-a estimation, OD at 750 nm (OD_750_) for cell density, instantaneous pigment fluorescence emission (FE) after being excited with 450 nm blue light (FE_450_) for chlorophyll-a or 620 nm red light (FE_620_) for phycocyanin, and quantum yield (QY) for photosynthesis efficiency of photosystem II. The results indicate that the acute toxicity was strain-, NMNP type-, dosage-, and endpoint-dependent. The two benthic strains *Microcoleus autumnale* and *Lyngbya* sp. were more resistant to NMNP treatment than the other seven free-floating strains. SWCNTs and fraction A14 of CND-G were more toxic than CND-G and CND-C/M. The CPP was the least toxic. The high dose generally caused more severe impairment than the low dose. OD_750_ and OD_680_ were more sensitive than FE_450_ and FE_620_. QY was the least sensitive endpoint. The strain dependence of toxicity suggested the potential application of these NMNPs as a target-specific tool for mitigating harmful cyanobacterial blooms.

## 1. Introduction

With the rapid changes in climate occurring over the past four decades, harmful cyanobacteria blooms (HCBs) have become a global environmental challenge, as cyanobacteria proliferate in warm, light-filled, and nutrient-abundant conditions [1]. If cyanobacterial populations are not appropriately controlled, they can quickly grow exponentially and form a bloom in their aquatic habitats. HCBs are of concern due to their ability to produce an array of toxins that may be harmful to environmental quality and human health [2]. Previously, scientists attempted a variety of chemical, biological, and mechanical approaches to mitigate cyanobacteria [3,4,5], e.g., the use of chemicals, metallic coagulants, flocculants, dredging, biological manipulation, and nutrient control. Although these approaches have proven to be effective, their short longevity, cytotoxicity to other non-target living organisms, and difficulty in application scale-up have motivated researchers to search for other methods of cyanobacterial mitigation.

Nanoparticles (NPs) are an intensively researched topic due to their diverse applications in such fields as medicine and pharmacy, electronics, agriculture, chemical catalysis, food industry, and so on [6]. Their nano-scale microscopic size and ability for modification give rise to many unique properties like photoactivity, cell surface adhesion, cell membrane penetration, and ability for cellular uptake [7,8,9]. The use of nanoparticles as an HCB control strategy has been pursued owing to their low cost of synthesis, environmental sustainability, and unique properties that could be exploited to specifically target cyanobacteria cellular pathways [10,11]. We are particularly interested in three classes of non-metallic nanoparticles (NMNPs), i.e., carbon nanodots (CNDs), carbon nanotubes (CNTs), and cell-penetrating peptides (CPPs), on their potential application to HCB mitigation.

CNDs and CNTs can be synthesized using simple, low-cost, green methods [12] and possess light-activated antimicrobial properties [13] regulated by their surface functionalization [14,15]. CNTs may be classified into two primary groups—multi-walled CNTs (MWCNTs) and single-walled CNTs (SWCNTs)—which are made of multiple layers and a single layer of carbon sheets, respectively. It is generally accepted that SWCNTs are more toxic than MWCNTs [16,17,18] due to their smaller size and higher surface area, allowing for greater interaction with biological systems [19]. CPPs are short peptides with a net positive charge capable of penetrating cell membranes in vivo and in vitro, without causing significant structural damage to the cell [9]. CPPs have been commonly used to deliver biologically active molecule cargos into cells with high efficiency and low toxicity [9,20]. Although various toxicological studies have been conducted on CNDs, CNTs and CPPs, their cyanobacterial toxicity—especially strain-specific cytotoxicity— is largely unknown. For instance, the pulmonary, reproductive, and developmental toxicities of CNTs to animals are well documented [19], but their impact on unicellular eukaryotic microalgae [21,22] and prokaryotic cyanobacteria [23] has not been thoroughly investigated. Thus, we launched the present study to compare the toxicity of CNDs, SWCNTs and CPPs to nine cyanobacterial strains belonging to seven genera (see Figure 1 for culture images). Two of these strains, *Microcoleus autumnale* CAWBG635 ATX and *Lyngbya* sp. CCAP 1446/10, are benthic species that often form mats at the bottom of natural water bodies and chunks of precipitates in flasks (see Appendix A).

## 2. Results

### 2.1. Experimental Setup and Control Treatments

We prepared and characterized in-house a batch of bPEI (branched polyethyleneimine)-functionalized SWCNT, two batches of CNDs (i.e., CND-G and CND-C/M), and a fraction (A14) from CND-G (named CND-G-A14) for cyanobacterial toxicity testing. CND-G and CND-C/M were prepared using bPEI along with glucose (CND-G) or chloroform/methanol (CND-C/M). We also purchased a 44-mer fusion CPP (γ-Zein-CADY) chemically synthesized by a commercial manufacturer. More details about the synthesis, purification, fractionation, and characterization of CNDs and CNTs, as well as more information about γ-Zein-CADY, can be found in Parts 2 and 3 of the Appendix A and Section 5.2 to Section 5.4 in Materials and Methods. Each of the nine cyanobacterial strains (Figure 1) were subjected to 48-h exposure to two concentrations of NPs, i.e., high and low, with high = 2 × low. Table 1 shows the experimental set-up. There were a total of 10 treatments plus the controls. Three experiments were conducted to test CND-G, CND-C/M, and CPP in the first, CND-G-A14 in the second, and SWCNT in the third. Each experiment had a separate control group. Each treatment (including the control) had three replicates of 5-mL cyanobacterial culture and 1~56 µL nanoparticles in 25-mL flasks.

The measurement results of the three sets of control treatments are shown in Figure 2. The measurement endpoints included cell density measured as optical density (OD) or light absorbance at a wavelength of 750 nm (OD_750_), chlorophyll-a (chl-a) content as OD at 680 nm (OD_680_), and red fluorescence emitted after being excited with blue light at a wavelength of 450 nm (FE_450_), phycocyanin as the emitted fluorescence intensity resulted from excitation with red light at a wavelength of 620 nm (FE_620_), and quantum yield (QY) (see Section 5.6 in Materials and Methods for more details).

There existed a statistically significant difference between the three sets of controls for three to seven tested strains, depending on the measurement endpoints (Figure 2). Even though all nine strains had approximately the same cell density (measured as OD_680_ = ~0.3) at the onset of each experiment, their readings of measurement endpoints differed greatly at the end of the experiment. For instance, the *M. autumnale* strain displayed the lowest average readings, whereas strains like *Synechocystis* sp. had the highest ones for all five endpoints. These results indicate that (a) differential growth rates for different strains, even among the three strains of the same *M. aeruginosa* species, and (b) the cell cultures of the same strain were not synchronized when prepared for the three sets of testing. Therefore, NP effects on each strain should be evaluated against the specific control group of the specific set of toxicity testing. In the following, we present the results of NP effects on nine strains by measurement endpoints.

### 2.2. Effects on Cell Density or Biomass Measured as OD_750_

OD_750_ measures the turbidity of cell cultures as most cell pigments exhibit very low absorbance at the 750 nm wavelength [24] and a culture becomes more turbid as the number of cells or cell density increases [25,26]. Since it is independent of photosynthetic activity, it can be used as an indicator of biomass. Here we compared OD_750_ measurements between the control and the treated to estimate effects of CND, SWCNT, and CPP on cell growth. As shown in Figure 3, all treatments, regardless of NP type and concentration, inhibited all strains except the benthic *Microcoleus autumnale*. Such inhibition was statistically significant, more often at the high dose than at the low dose, for all nanoparticles. Among the nine strains, *Synechocystis* sp. PCC 6803 was the most vulnerable, as its OD_750_ showed statistically significant decreases in response to acute exposure to all nanoparticles at both high and low doses (*p* < 0.01 for CND-C/M low dose and SWCNT high dose; *p* < 0.001 for all other treatments). The second- and third-most vulnerable strains were *Planktothrix agardhii* (*p* < 0.001 for six treatments and *p* < 0.01 for CND-C/M high dose) and *Aphanizomenon* sp. (*p* < 0.001 for two treatments, *p* < 0.01 for two treatments, and *p* < 0.05 for three treatments), respectively. The two benthic strains were the least vulnerable as the observed effects were largely insignificant. *M. autumnale* even showed a statistically significant growth increase of 329% and 29% in the CND-G-A14 high dose (*p* < 0.001) and CPP high dose (*p* < 0.01) treatments, respectively.

The tested nanoparticles showed differential effects on cell growth. Generally speaking, SWCNT and the A14 fraction of CND-G were more toxic than CND-G, CND-C/M, and CPP. Specifically, CND-G-A14 had statistically significant inhibitory effects on the biomass of all tested strains with the exception of two benthic strains. This might be due to the fact that the applied CND-G-A14 doses (both high and low) were 2.36, 2.57, and 2.66 times as their counterparts of CND-G, CND-C/M, and SWCNT, respectively. If taking the dose difference into consideration, SWCNT was the most toxic as it showed significant adverse impacts (*p* < 0.05) on six of the seven non-benthic strains. CPP was apparently the least toxic with statistically significant but less severe effects on five of the seven non-benthic strains, mostly at the high dose, despite that the applied CPP doses were 2.5 times those of CND-G-A14 or roughly 6.25 times those of CND-G, CND-C/M, and SWCNT.

### 2.3. Effects on chl-a Content Measured as OD_680_ and FE_450_

As the primary light-harvesting photosynthetic pigment, cyanobacterial chl-a has a peak absorbance at 680 nm and can emit red fluorescence after excitation at 450 nm. Hence, both OD_680_ [24] and FE_450_ [27,28] can measure the chl-a content in the cyanobacterial cell culture. The OD_680_ and FE_450_ results shown in Figure 4 and Figure 5 are very similar to the OD_750_ results shown in Figure 3, because photosynthetic cyanobacteria use chl-a as the major pigment for photosynthesis, making this endpoint a good parameter indicative of photosynthetic activity and population growth. The similarities in the testing results between the three endpoints (OD_750_, OD_680_ and FE_450_) are reflected in not only the overall dose-response pattern across all NPs and strains but also strain-specific and nanoparticle-specific patterns. For instance, all treatments caused inhibitory effects on all three endpoints, with *M. autumnale* being the only exception. Based on the degree and statistical significance of impacts on chl-a content, *Synechocystis* sp. was the most affected strain, followed by *P. agardhii*, *Aphanizomenon* sp. and *M. aeruginosa* LE3. When evaluated using OD_680_ and FE_450_, CND-G-A14 and SWCNT showed more pronounced effects than CND-G and CND-C/M, while CPP was the least toxic, all consistent with the outcome using OD_750_.

Despite high similarities, there are also some noticeable differences between the three endpoints. For instance, the effects of SWCNT on *M. aeruginosa* UTEX 2386 were statistically insignificant when assessed using OD_750_ (biomass) but statistically significant when assessed using either OD_680_ or FE_450_ (chl-a). In response to CND-G-A14 high dose and CPP high dose treatments, the OD_680_/FE_450_ of *M. autumnale* increased by 600%/338% and 24%/46%, respectively, all of which were statistically insignificant, albeit similar to or even higher than the degree of alteration for OD_750_ (329% and 29%, *p* < 0.001 and 0.01, respectively). On the other hand, FE_450_ showed fewer statistically significant events caused by NPs (see details broken down by strain in Appendix A or by NP class in Appendix A) than did OD_750_ and OD_680_, generally on all nine cyanobacterial strains and specifically on *Synechocystis* sp., *Aphanizomenon* sp., and *P. agardhii*. These results suggest that FE_450_ is less sensitive than OD_680_ and OD_750_. This might be due to the chl-a fluorescence (signal peaked at 680 nm) being interfered by phycocyanin fluorescence (peaked at 640 nm), as both pigments are present in cyanobacteria [29]. On the other hand, optical density is the most accurate as proxy measurements for biomass (OD_750_) and pigment (chl-a) concentration (OD_680_) in homogenous single-phase cell cultures where light attenuation is dominated by absorbance [30]. However, OD measurements may not be accurate in heterogenous cultures of benthic strains (e.g., *Microcoleus* sp. and *Lyngbya* sp.), where light attenuation may also be attributed to scattering due to difference in refractive index and the shape of this index mismatch in multi-phased suspensions [31].

### 2.4. Effects on Phycocyanin Measured as FE_620_

FE_620_ measures the concentration of phycocyanin, a phycobilisome pigment unique to freshwater cyanobacteria [32]. As shown in Figure 6, FE_620_ measurements were suppressed by NP treatments in 68% (61/90) cases and stimulated in 29% (26/90) cases, with three cases where no effect (i.e., 0% alteration on average) was observed. One strain treated with one NP dose was considered an individual case; hence, there were 90 cases (=9 strains × 5 NPs × 2 doses) in total. Statistically significant suppression and stimulation occurred in twenty-one and four cases, separately (Appendix A). The overall alteration degree (Appendix A) and total number of statistically significant cases (Appendix A) were both lower with FE_620_ than with FE_450_ and OD_680_, suggesting that phycocyanin might be less affected than chl-a by NP treatment.

Strain- and NP-dependent sensitivity was also observed with the FE_620_ endpoint (Figure 6, Appendix A). Similar to the outcome of OD_750_, OD_680_ and FE_450_ measurements, the two benthic strains displayed the highest resistance to NPs whereas *Synechocystis* sp. and *P. agardhii* were the most vulnerable. Another similarity between FE_620_ and FE_450_ is that *Aphanizomenon* sp. showed much less sensitivity than when it was evaluated by OD_750_ and OD_680_. There existed a notable difference for the three *M. aeruginosa* strains: the LE3 strain exhibited similar sensitivity as the UTEX 2386 and 2385 strains when evaluated by FE_620_ but higher sensitivity when evaluated by OD_750_, OD_680_, and FE_450_. NPs showed a similar pattern of differential toxicity as evaluated by all four endpoints: SWCNT > CND-G-A14 > CND-G and CND-C/M > CPP. The high dose caused more pronounced effects than the low dose. However, the most toxic SWCNT had a lesser effect on phycocyanin production, as the two SWCNT doses caused no or less significant decreases on FE_620_ than on OD_750_, OD_680_ and FE_450_ in four non-benthic strains, including *Synechocystis* sp., *M. aeruginosa* UTEX 2385, *Aphanizomenon* sp., and *P*. *agardhii*.

### 2.5. Effects on Photosynthetic Efficiency Measured as QY

Quantum yield (QY) is an important parameter for assessing the health of photosystem II (PS II) in cyanobacteria [27]. Statistical data analysis indicates that only SWCNT significantly inhibited the PS II photosynthetic efficiency in all strains except *M. autumnale* (Figure 7). Other NPs showed little significant effects on any of the nine strains with only four exceptions (i.e., CND-G-A14 on *P. agardhii* (both high and low doses) and *Aphanizomenon* sp. (high dose), and CPP on *M. aeruginosa* LE3 (high dose)) (see also Appendix A). The total number of statistically significant cases determined by QY was 21 (including one case of stimulation), as compared to 41 (2), 44 (0), 31 (0), and 25 (4) determined by OD_750_, OD_680_, FE_450_, and FE_620_, respectively (Appendix A). These results suggest that QY might be the least sensitive endpoint among the five. However, one may gain some insights from the QY results into the potential toxicological mechanism of SWCNT, which is likely driven by damages to PS II in cyanobacteria.

## 3. Discussion

In the current study, we determined the acute cyanotoxicity of three types of NMNPs (SWCNTs, CNDs and CPPs) by measuring five physiological endpoints (OD_750_, OD_680_, FE_450_, FE_620_ and QY) of nine cyanobacterial strains after a 48-h exposure to two doses. The observed responses were endpoint-, strain-, NP type-, and NP dose-dependent. Generally speaking, the optical density-based endpoints (OD_750_ and OD_680_) were more sensitive than the fluorescence-based endpoints (FE_450_, FE_620_ and QY) with QY being the least sensitive. SWCNTs were more toxic than CNDs while CPPs were the least toxic. The higher dose exerted more pronounced effect than the lower dose, especially for the three CND preparations where the doses of CND-G-A14 were twice those of CND-G and CND-C/M. The nine test strains showed differential sensitivity to the NMNPs, with *Synechocystis* sp. being the most vulnerable and *M. autumnale* being the most tolerant. In the following, we discuss the differential sensitivity between different strains and endpoints as well as the differential cyanotoxicity of NMNPs and their toxicological mode of action.

### 3.1. Differential Sensitivity of Strains to NMNPs

The nine cyanobacterial strains treated with NMNPs represent a variety of cyanobacterial genera, each with specific morphological features that may be attributed to their differential sensitivity. For instance, the unicellular, spherical strains in the *Synechocystis* genus neither form cell aggregates nor develop protective sheath in laboratory culture [33]. Such structural features likely made the *Synechocystis* sp. PCC 6803 strain vulnerable to the toxic NMNPs. On the other hand, the benthic genera *Microcoleus* (formerly *Phormidium*) and *Lyngbya* (both belonging to the order Oscillatoriales) often form mats of many centimeters thick in the environment [34]. The two Oscillatoriales strains used in this study were observed to form thick cell clumps that stuck to the culture flask under laboratory conditions for axenic culturing as shown in Appendix A. They also organize multiple trichomes in firm sheath and form multi-layered or three-dimensional sheaths that are very thick to lyse or penetrate [33,35]. Therefore, such unique cellular morphology might enable them to exhibit the highest resistance to NMNPs in this study.

Although the three *M. aeruginosa* strains are similar to *Synechocystis* sp. in their spherical cell morphology, they showed a higher tolerance, probably owing to their possession of a physical defense mechanism through excreting a mucilage layer [35,36]. While *A. cylindrica*, *Planktothrix* sp., and *Aphanizomenon* sp. are filamentous like the two benthic strains, but they do not form as thick of sheaths like *Lyngbya* and *Microcoleus*. Nevertheless, they can form thin mucilaginous layers in unfavorable conditions [33,37]. Therefore, these six strains displayed intermediate resistance to NMNPs, i.e., more resistant than *Synechocystis* but less resistant than *Lyngbya* and *Microcoleus*.

Extracellular polysaccharides or exopolysaccharides (EPS) excreted by cyanobacteria plays a major role in protecting cells from stress in extreme habitats and from other harmful conditions such as antibacterial agents [38]. However, EPS vary greatly in their amount, molecular weight (MW) and composition between different cyanobacterial species [39]. It was reported that the higher MW and content of EPS released by a *Microcoleus* sp. had positive impacts on its growth and resistance to stress when compared with two other species belonging to the order Nostocales [39]. Chen et al. [40] observed that a colonial *M. aeruginosa* strain CHAOHU 1326 produced a higher amount of EPS than other three unicellular *M. aeruginosa* strains. The authors further elucidated the genetic basis for colonial aggregation of CHAOHU 1326, which was active transcription of an 18-gene cluster responsible for EPS biosynthesis. These studies imply the role of EPS in bloom formation and resistance to the toxicants produced during HCB events. It would be worth further examination to investigate the relationship between EPS production or EPS-synthesis genes activities and resistance to NMNP in the nine test cyanobacterial strains.

### 3.2. Differential Sensitivity of Measurement Endpoints

Our acute 48-h toxicity testing results show that measurement endpoints related to fluorescence emission of pigments (e.g., FE_450_ for chl-a and FE_620_ for phycocyanin) were less sensitive than cell density or biomass measurement endpoints (e.g., OD_750_ and OD_680_). This might be due to the delay in degradation of pigments released from damaged cells. The FE_450_ and FE_620_ measurements might account for both extracellular and intracellular chl-a and phycocyanin, as previously reported by Keliri et al. [28], leading to the underestimation of cell damages caused by NMNP treatments. For example, some of the strains such as *P. agardhii* and *M. aeruginosa* LE3 showed significant increases in FE_620_ compared to the control. This could be due to extracellular phycocyanin because of cell damage.

Photosystem II (PSII) in cyanobacteria is a protein complex embedded in the thylakoid membrane that uses light energy to split water molecules, producing oxygen and transferring electrons [41]. Key pigments involved in this process include chl-a, phycoerythrin, and phycocyanin, with chl-a being the primary light-absorbing pigment within the reaction center of PSII; these pigments act as an antenna system to capture light energy and transfer it to the reaction center for electron transfer reactions. The AquaPen device used for QY measurements only accounted for chl-a-mediated photosynthetic efficiency of PS II. Our results show that only SWCNT significantly inhibited QY, suggesting that SWCNT might act directly on the chl-a pigment or other components involved in the photosynthetic activities of PS II. The CNDs and CPPs did not cause significant effects on QY (with only a few exceptions), indicating that they may have a mode of action distinctly different from SWCNT.

### 3.3. Toxicological Mode of Action

Many researchers in the past have attempted to mitigate cyanobacterial blooms using primarily metallic nanoparticles (see reviews [6,10,11,42]). As summarized in recent reviews [10,11,43], the main mechanisms of nanotoxicity to cyanobacteria include the following: (1) shading on cyanobacterial cell surface reduces light availability; (2) ultra-structural damage to cell membrane, cell wall, and intracellular organelles; (3) photo-catalyzing- or oxidative stress-generated reactive oxygen species (ROS) lead to increases in activities of antioxidant enzymes and non-enzymatic components like proline and osmolytes; (4) pigment impairments lead to reduced photosynthesis and growth rate; and (5) flocculation- or coagulation-mediated precipitation or removal. Although nanotoxicity is dependent on many factors such as the size, shape, composition, surface area, coating, oxidation state, charge, aggregation state, crystalline phase, and solubility [44], most of the aforementioned mechanisms may be applicable to NMNPs. 

For instance, Yao et al. [45] conducted a toxicity study on the eukaryotic unicellular green algae *Scenedesmus obliquus* with carbon quantum dots (CQDs), a nanoparticle with semi-conductive properties and similar to CNDs in structure. The authors reported that CQD suspensions caused a significant growth inhibition in the microalgae, and that the toxicity mechanism of CQD suspensions was likely due to increased oxidative stress. Although this study used different organisms and nanoparticles, their results may still be applicable to the toxicological mechanism of CNDs tested in experiments due to the physicochemical and size similarity in the nanoparticles used. Abu Rabe et al. [15] provided further evidence on oxidative stress being the potential mechanism of CND toxicity to multi-drug resistant (MDR) bacteria. The CNDs were employed as a photodynamic inactivation (PDI) agent that generated ROS under light illumination and effectively controlled the MDR bacteria. In another study, the same research group tested antimicrobial properties of CNDs via light inactivation and determined that surface functionalization, specifically the addition of a positive functional group that releases protons, enabled a more effective antimicrobial activation [14].

Sohn et al. [21] reported the median effective concentration (EC_50_) of a commercial SWCNT to two freshwater microalgae (*Raphidocelis subcapitata* and *Chlorella vulgaris*) being 30 mg/L, which is comparable with the high dose of SWCNT (1.5 mg/L) tested in the current study. The authors observed that SWCNT inhibited the growth of microalgae via “flocculation”, which is similar to the clumping phenomena we observed in our SWCNT-treated cyanobacterial samples. Bennett et al. [8] investigated the toxicity mechanism of five commercial SWCNTs (10 mg/L) in treated microalgae (*Pseudokirchneriella subcapitata*) and determined that toxic effects were not caused by shading but instead by photoactivity that produced such ROS as singlet oxygen and superoxide. The authors also observed a considerable difference between purified and raw SWCNTs, with the latter showing little or no photoactivity. Therefore, it is necessary to characterize the toxicity and mode of action for each SWCNT individually.

While studying the mechanism of CPP (R9)-mediated delivery of a green fluorescent protein (GFP), Liu et al. [46] evaluated the cytotoxicity of R9/GFP (2400 nM/800 nM) complex to *Synechocystis* sp. PCC 6803 and observed no effect on cell viability. Kilk et al. [47] assessed the in vitro toxicity of five CPPs to a Chinese hamster ovary (CHO) cell line with a particular focus on their metabolomic profiles. Among the five CPPs, transportan at >5 µM caused unreversible alteration of metabolome, including cellular redox potential, depleted energy and the pools of purines and pyrimidines. Although these studies investigated different CPPs or different cell lines, the CPP concentrations are comparable to the high dose (10 mg/L or 2 µM) of γ-Zein-CADY used in current study. Taken together, all the results suggest low or little cytotoxicity of CPPs, but further research is warranted to investigate sublethal effects at the molecular level.

### 3.4. Future Research Directions

There are many other phenotypic and genetic/genomic endpoints to evaluate the toxicity of and gain mechanistic insights into NMNPs in cyanobacteria besides those determined in the present study. For instance, scanning electron microscopy (SEM) can be used to visualize the cell surfaces and observe any physical damage to cyanobacterial cells that could further elucidate the degree and mechanism of toxicity [40]. “Omics” research is also of interest in studies of toxicity, as the changes in cells exposed to substances can be measured at the gene expression level, and the mechanism of toxicity can be characterized by looking at differences in genetic biomarkers. Molecular studies can also be species-specific, which would help determine why strains of different genotypes react differently to the same toxicant. Improved methods of measuring toxicity levels could better clarify the toxicity of NMNPs to cyanobacteria and defense mechanisms of different genera.

It is also worth noting that the present experiments took place in a controlled laboratory setting, and factors such as environmental stress or the effects of other aquatic organisms (e.g., other phytoplanktonic, zooplanktonic, and even predatory fish species) were not taken into account in this study. In one study, growth inhibition caused by the dispersal of SWCNTs in aquatic conditions was highly correlated with the shading and agglomeration that the algae were experiencing in the environment; thus, nanoparticle toxicity was related to indirect effects that would be highly variable depending on environmental conditions such as vertical mixing [48]. There have been no studies reporting the effects of nanoparticles in cocultures of multiple cyanobacterial strains or a liquid medium comprised of cyanobacteria and other planktonic species like green algae [42]. The joint toxicities of multiple classes of NMNPs (e.g., CNDs and CNTs) have rarely been studied. The sole documented study reported enhanced adverse effects of CNDs, when paired with CNTs, on cyanobacteria due to improved photocatalytic properties and efficiency through an increase of electron transfers between the two classes of NMNPs [49]. An implication of these results is that we may further explore beneficial coupling of different NMNP to strengthen their efficacy in mitigating cyanobacterial blooms.

The cyanotoxicity testing protocol employed in the present study aligned in principle with the standard procedure of 96-h algal toxicity testing published by US EPA [50], ASTM [51] and OECD [52]. The acute toxicity data collected for NMNPs to pure cultured cyanobacterial strains may be extrapolated to predict long-term ecological effects if full consideration can be given to a number of deterministic factors that could alter toxicity profiles [53]. These factors include, but are not limited to, nanoparticle persistence, aggregation, and interactions with biotic (e.g., uptake, biodegradation, bioaccumulation) and abiotic (e.g., pH, redox, temperature) variables.

The present study was limited to toxicity evaluation without looking into the cyanotoxins produced by NMNP-exposed cyanobacteria. It is well documented that many harmful algal bloom-forming cyanobacterial species may produce a wide variety of cyanotoxins (e.g., anatoxin, microcystin, saxitoxin; see https://www.epa.gov/habs/learn-about-harmful-algae-cyanobacteria-and-cyanotoxins (accessed on 18 March 2025)) [2,34]. It remains largely unexplored whether the observed cellular stress or lysis induced by NMNP exposure might trigger the release of intracellular cyanotoxins into the water bodies. Future research is warranted to investigate and compare NMNP treatments with existing copper- and hydrogen peroxide-based algaecides in terms of cost-effectiveness and environmental sustainability (e.g., cyanotoxin release and human health impacts).

## 4. Conclusions

This study is the first of its kind in terms of reporting the acute toxicity of five non-metallic nanoparticle (NMNP) preparations to a wide range of cyanobacterial strains representing seven commonly found genera and both benthic and floating species. Our goal was to determine the potential of SWCNTs, CNDs, and CPPs in mitigating harmful bloom-forming cyanobacteria in aquatic systems. We demonstrated that 48-h exposure to two doses of these NMNPs caused strain-, endpoint-, and chemical-dependent adverse effects on the treated axenic cultures of nine cyanobacterial strains. *Synechocystis* sp. PCC 6803 was the most sensitive while the benthic strain *M. autumnale* was the most resistant. OD_750_ and OD_680_ (the endpoints for cell density and biomass estimation) were more sensitive than FE_450_ (chl-a) and FE620 (phycocyanin), while QY (PSII photosynthetic efficiency) was the least sensitive endpoint. The in-house synthesized SWCNT was the most toxic, whereas the CPP was the least toxic. To further explore the application of NMNPs to HABs mitigation, full-scale dose-response studies are required to characterize the lowest observable effective dose (LOED) and the effective dose causing 50% inhibition (ED_50_) for each NMNP. More in-depth studies are also warranted to better understand their largely unknown mode of actions and impacts on cyanotoxin production and investigate how environmental factors may modify their cyanotoxicity.

## 5. Materials and Methods

### 5.1. Cyanobacteria Cultures

Nine laboratory axenic cultures of cyanobacterial strains were used. Three *Microcystis aeruginosa strains* (UTEX 2386, UTEX 2385, and LE3), *Lyngbya* sp. CCAP 1446/10, and *Synechocystis* sp. PCC 6803 were maintained in 1× Cyanobacteria BG-11 Freshwater solution (Millipore Sigma, St. Louis, MO, USA), *Anabaena cylindrica* PCC 7122, *Planktothrix agardhii* SB 1810, and *Aphanizomenon* sp. NZ in Jaworski’s (JM) Medium [54], and *Microcoleus autumnale* CAWBG635 ATX in MLA Medium [55]. Ten days before the toxicity testing, cultures were freshly inoculated in 100 mL of their respective medium in 250 mL glass flasks. Then, the cultures were grown under a 12/12 (light/dark) cycle and a set temperature of 21 °C with constant shaking at 85 rpm in an Algae Growth Chamber Model AL-41L4 (Percival Scientific, Perry, IA, USA). Optical density (OD) of cyanobacterial cultures was measured using a Shimadzu UV 1800 Spectrophotometer (Columbia, MD, USA) as absorbance at 680 nm (OD_680_). Prior to toxicity testing, the OD_680_ of cultures was adjusted to 0.3 by dilution using their respective growth media.

### 5.2. Cell-Penetrating Peptide (CPP)

γ-Zein-CADY, a 44 mer fusion peptide consisting of two CPPs, an 18 mer γ-zein (amino acid sequence: VRLPPPVRLPPPVRLPPP) and a 19 mer CADY (amino acid sequence: GLWRALWRLLRSLWRLLWK), and a 5 mer endosomal escape domain (EED, amino acid sequence: GFWFG) connected by two spacers (amino acid glycine or G per spacer), was designed in-house by following the scheme “CPP1 (γ-zein)-Spacer-CPP2 (CADY)-Spacer-EED”. The fusion CPP γ-zein-CADY (molecular weight = 5211) was chemically synthesized and purified using HPLC by ABI Scientific (Sterling, VA, USA) with a purity of >95%, a Z-average (intensity weighted harmonic mean of particle size distribution) of 125.2 nm and a zeta-potential of 10.3 mV. Polypeptides like γ-Zein-CADY spontaneously self-assemble into nanoparticles, so no stable structure could be inferred [56].

### 5.3. Carbon Nanodot (CNDs)

Two methods described in Schwartz et al. [7] were adopted with modification to prepare in-house three batches of CNDs, one synthesized from 10-kDa branched polyethyleneimine (bPEI 10000, molecular weight = 10,000, purity = 99%; Polysciences, Warrington, PA, USA) and the solvent chloroform: methanol (4:1) and two from bPEI and glucose. The two CND synthesis methods are briefly described below. More details about the synthesis, purification, fractionation and characterization can be found in Appendix A.

#### 5.3.1. CND-G Synthesis Using 10-kDa bPEI and Glucose

For the synthesis of PEI functionalized glucose carbon dots, a 5 mL solution containing 37.5 mg/mL glucose and 75 mg/mL 10-kDa bPEI was adjusted to pH 8.0 with 4N HCl and degassed under vacuum. The solution was heated in Monowave from 50 to 100 °C over 3 min and then held at 100 °C for an additional 5 min. Two batches of CND-G were synthesized. The first batch was purified and fractionated using a BioRad NGC Quest 10 system. Nine collected fractions (A9 to A17) were further characterized for their absorbance maxima (360 nm, Appendix A) and particle size distribution. Each fraction contained both large (500–1200 nm in diameter) and small (5–9 nm in diameter) particles (Appendix A). The second batch was fractionated using a BioRad BioLogic DuoFlow FPLC system. Fractions 37–71 were consolidated and designated as CND-G with a Z-average of 7–10 nm (Appendix A). Fraction A14 from the first batch (designated as CND-G-A14, Z-average = 279 nm, Appendix A) and the second batch CND-G were chosen for cyanobacterial toxicity testing. See Appendix A for CND-G molecular structure.

#### 5.3.2. CND-C/M Synthesis Using 10-kDa bPEI and CHCl_3_/CH_3_OH

The bPEI 10000 (375 mg) was added to 5 mL of the chloroform: methanol solvent, which was heated to 155 °C in a microwave reactor (Monowave 50, Anton Paar GmbH, Austria) with a ramp time of 5 min and held at 155 °C for an additional 7 min. The reaction products were dried under nitrogen and then resuspended in water. Any residual chloroform was separated from the aqueous solution of carbon dots by centrifugation at 7500 g for 5 min. The pH of the carbon dot solution was adjusted to 8.0 with 4N HCl and the final volume was adjusted to 5 mL with water. The prepared solution was further purified and fractionated using the same BioRad BioLogic DuoFlow FPLC system as for the second CND-G batch. Fractions 44 to 71 were consolidated into one preparation designated as CND-C/M with a Z-average of 7–8 nm in diameter and a maximal absorbance at 345 nm. See Appendix A for more info. This CND-C/M batch was used for cyanobacterial toxicity testing. See Appendix A for CND-C/M molecular structure.

### 5.4. Single-Walled Carbon Nanotubes (SWCNTs)

The procedure used in this study was adapted from Demirer et al. [57] with some modifications to synthesize bPEI functionalized SWCNT. See SM Part 3 for detailed descriptions. Briefly, the carboxyl (COOH) functionalized SWCNTs were first suspended in nuclease free water. Then, coupling regents (EDC and NHS) were introduced to activate the SWCNT-COOH suspension. Next, bPEI was grafted onto the SWCNT via amide bond formations with the utilization of coupling agents, and the solution was washed multiple times to remove any unreacted coupling agents or unbound bPEI. Finally, bPEI grafted SWCNTs were re-suspended in nuclease-free water to be used in toxicity studies. Three batches of PEI-SWCNT were prepared. The concentration of PEI-SWNTs was measured via absorbance at 632 nm with an extinction coefficient of 0.036 L mg^−1^ cm^−1^. Batch 1A was chosen for cyanobacterial toxicity testing. It had a concentration of 138 mg/L (Appendix A), a Z-average of 153 nm (Appendix A) and an average Zeta potential of 62.7 ± 7.56 (Appendix A). See Appendix A for SWCNT molecular structure.

### 5.5. Cyanobacterial Toxicity Testing Protocol

Each of the nine cyanobacterial strains in the lab was tested with all five nanoparticle preparations at two concentrations (high and low with high = 2 × low). At the onset of toxicity testing, the OD_680_ value of all nine cyanobacterial cultures was set at approximately 0.3, and the nanoparticle (NP)-treated cultures were returned to the growth chamber under the same conditions, i.e., 12 hrs (light)/12 hrs (dark) cycle and temperature set at 21 ± 1 °C with constant shaking at 85 rpm. After 48-h exposure, photos were taken of the flasks or 6-well plates containing the transferred SWCNT-treated culture (see Appendix A). One mL of culture from the flask was pipetted into a cuvette for endpoint measurements.

### 5.6. Data Collection of Endpoint Measurements

Besides OD_680_, absorbance at 750 nm (OD_750_) was also determined using a UV-1800 Spectrophotometer (Shimadzu, Japan). Instantaneous fluorescence emission was measured using an AquaPen AP-C 100 (Photon Systems Instruments, Drasov, Czech Republic) with a blue excitation wavelength at 450 nm (FE_450_) and or a red excitation wavelength at 620 nm (FE_620_), in addition to quantum yield (QY). FE_450_ is proportional to the chl-a content, whereas FE_620_ is proportional to the phycocyanin concentration [28]. The QY of photosynthesis system II (PSII) is a measure of the emitted variable chlorophyll fluorescence after a series of light pulses and calculated as the ratio of the variable fluorescence (Fv) to the maximum fluorescence intensity in the dark-adapted state (Fm) [27,28]. Percent changes between the mean value of the control group and those of the NP treatment groups and standard deviations for each strain and each measurement endpoint were reported.

### 5.7. Data Statistical Analysis

To determine if NP treatments had a statistically significant effect, a one-way analysis of variance (ANOVA) was performed for each strain and each measurement endpoint, followed by a post hoc Dunnett’s test. All analyses were performed in the R software environment for statistical computing and graphics [58]. The one-way ANOVA was performed using the R Stats package with levels of significance set at confidence intervals of 95%, 99%, and 99.9%. A post hoc Dunnett test was conducted to compare every treatment mean to the control mean using the DescTools package [59]. The statistical difference between the three sets of controls was analysed using ANOVA with Tukey HSD and significance set at *p* < 0.05.

## Figures and Tables

**Figure 1 toxins-17-00172-f001:**
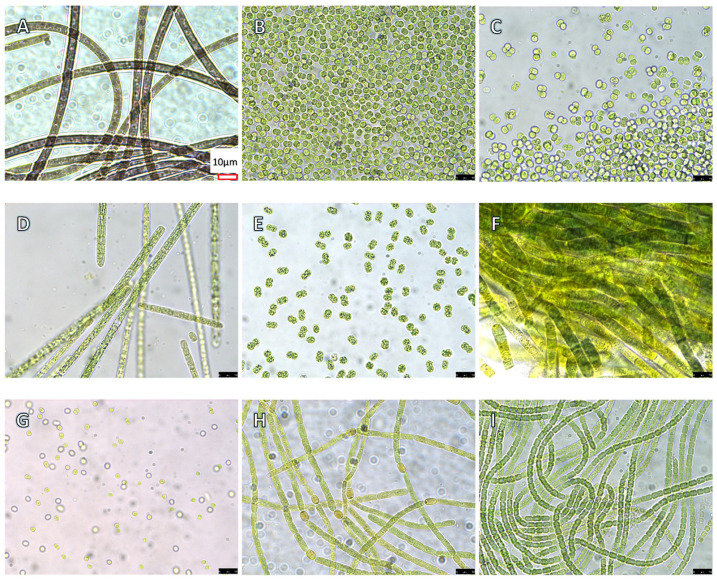
Microscopic images of nine axenic cyanobacterial cultures used in the present study at a 1000× magnification. The scale bar is redrawn and inserted at the bottom right corner in panel A for sharing by all nine panels due to blurriness of the original scale bars. (**A**) *Microcoleus autumnale* CAWBG635 ATX; (**B**) *Microcystis aeruginosa* UTEX 2385; (**C**) *M. aeruginosa* UTEX 2386; (**D**) *Planktothrix agardhii* SB 1810; (**E**) *M. aeruginosa* LE3; (**F**) *Lyngbya* sp. CCAP 1446/10; (**G**) *Synechocystis* sp. PCC 6803; (**H**) *Aphanizomenon* sp. NZ; and (**I**) *Anabaena cylindrica* PCC 7122.

**Figure 2 toxins-17-00172-f002:**
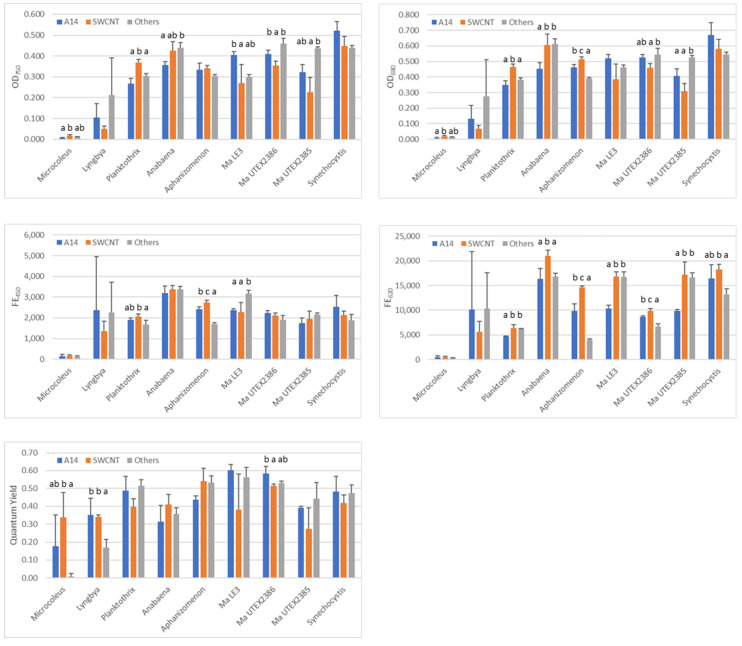
Measurements of five endpoints at the end of 48-h testing in the control treatment for nine cyanobacterial strains. Three sets of controls were prepared for testing CND-G-A14 (A14), SWCNT or other NPs (Others; including CND-G, CND-C/M, and CPP). The three Microcystis aeruginosa strains are labeled as Ma LE3, Ma UTEX2386 and Ma UTEX2385. Shown are average (column) and standard deviation (error bar) with n = 3 for each individual strain in each set of controls. Statistically significant differences between the three sets of controls (ANOVA with Tukey HSD, *p* < 0.05) are indicated with different letters. If a strain has no letter shown, there existed no significant difference between the three sets of controls.

**Figure 3 toxins-17-00172-f003:**
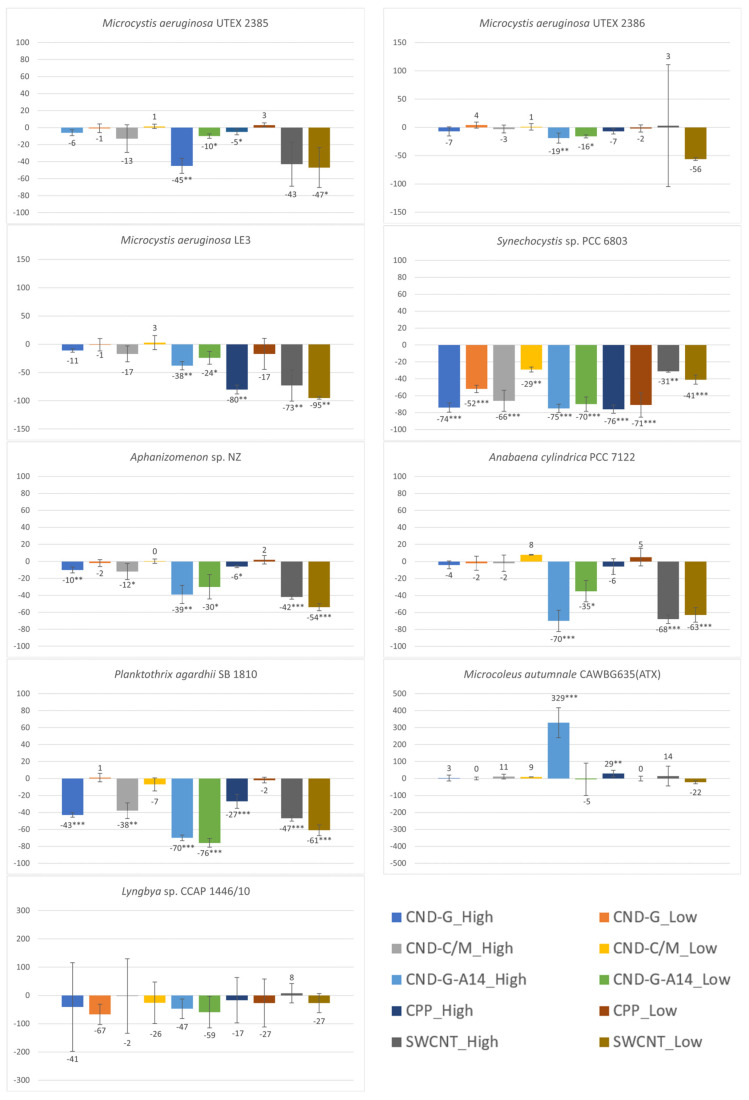
Effects of 48-h exposure to synthesized CND, SWCNT, and CPP on the cell density of nine cyanobacterial strains measured as OD_750_. Effects are expressed as percentage change in OD_750_ of treated samples over that of untreated controls. Statistical significance of ANOVA: * *p* < 0.05; ** *p* < 0.01; *** *p* < 0.001. Columns represent the average % change over the control as indicated by the numbers above or below the columns, whereas error bars represent standard deviation (n = 3).

**Figure 4 toxins-17-00172-f004:**
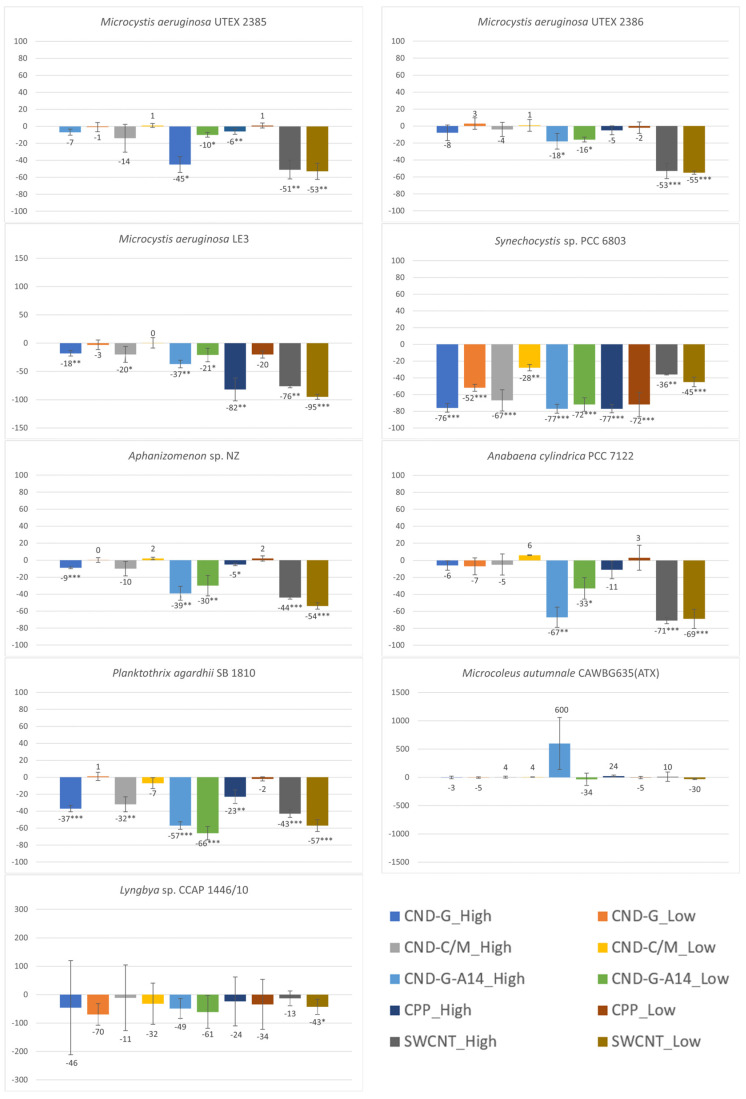
Effects of 48-h exposure to synthesized CND, SWCNT, and CPP on the chl-a content in nine cyanobacterial strains measured as OD_680_. Effects are expressed as percentage change in OD_680_ of treated samples over that of untreated controls. Statistical significance of ANOVA: * *p* < 0.05; ** *p* < 0.01; *** *p* < 0.001. Columns represent the average % change over the control as indicated by the numbers above or below the columns, whereas error bars represent standard deviation (n = 3).

**Figure 5 toxins-17-00172-f005:**
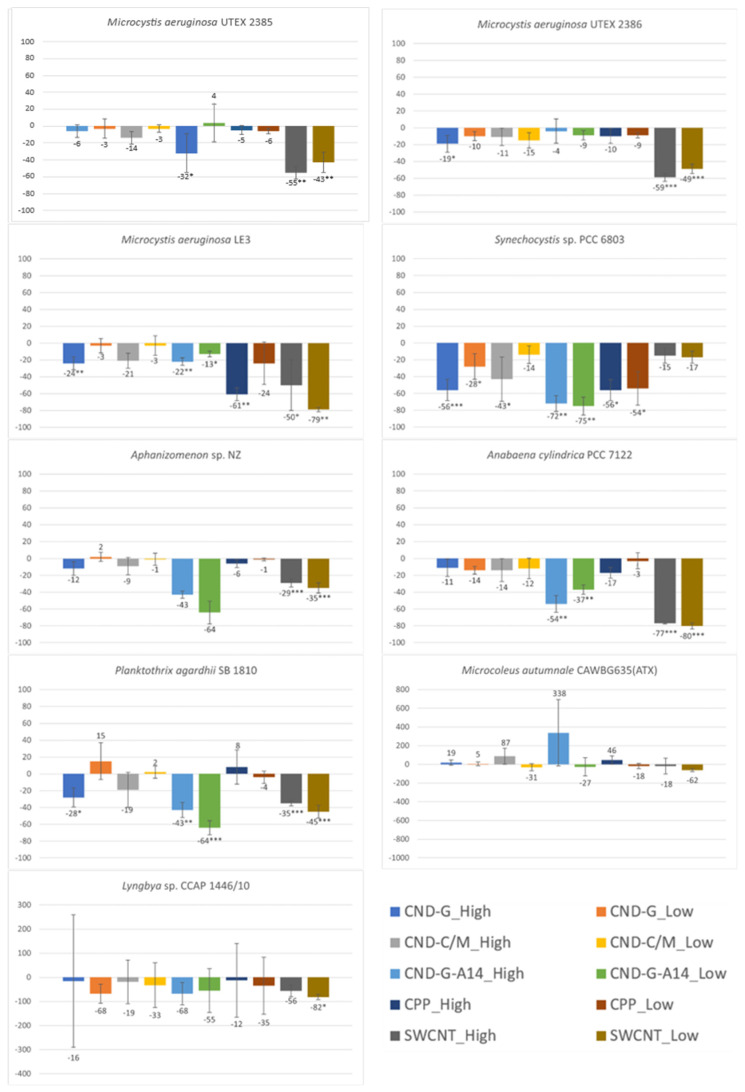
Effects of 48-h exposure to synthesized CND, SWCNT, and CPP on the chl-a content in nine cyanobacterial strains measured as FE_450_. Effects are expressed as percentage change in FE_450_ of treated samples over that of untreated controls. Statistical significance of ANOVA: * *p* < 0.05; ** *p* < 0.01; *** *p* < 0.001. Columns represent the average % change over the control as indicated by the numbers above or below the columns, whereas error bars represent standard deviation (n = 3).

**Figure 6 toxins-17-00172-f006:**
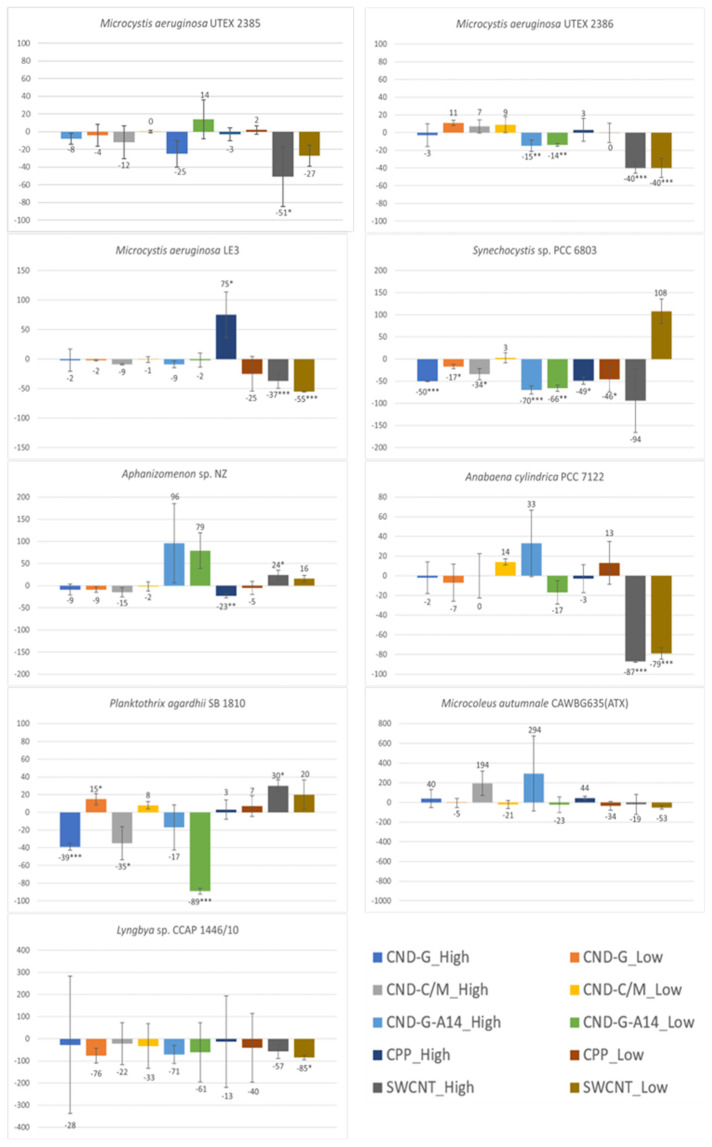
Effects of 48-h exposure to synthesized CND, SWCNT, and CPP on phycocyanin of nine cyanobacterial strains measured as FE_620_. Effects are expressed as percentage change in FE_620_ of treated samples over that of untreated controls. Statistical significance of ANOVA: * *p* < 0.05; ** *p* < 0.01; *** *p* < 0.001. Columns represent the average % change over the control as indicated by the numbers above or below the columns, whereas error bars represent standard deviation (n = 3).

**Figure 7 toxins-17-00172-f007:**
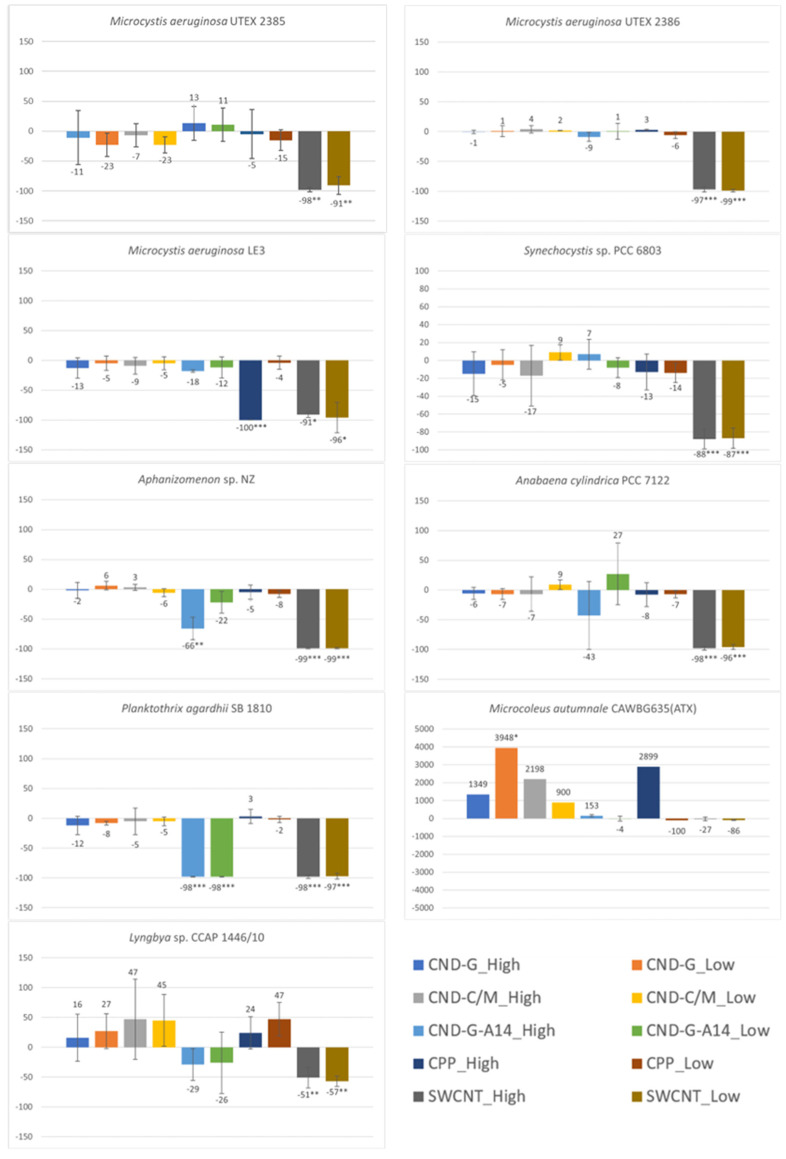
Effects of 48-h exposure to synthesized CND, SWCNT, and CPP on photosynthetic efficiency of nine cyanobacterial strains measured as QY. Effects are expressed as percentage change in QY of treated samples over that of untreated controls. Statistical significance of ANOVA: * *p* < 0.05; ** *p* < 0.01; *** *p* < 0.001. Columns represent the average % change over the control as indicated by the numbers above or below the columns, whereas error bars represent standard deviation (n = 3).

**Table 1 toxins-17-00172-t001:** Experimental set-up of the 48 hr acute cyanobacterial toxicity testing for five nanoparticles at two concentrations.

Treatment	Control	CND-G_High	CND-G_Low	CND-C/M_High	CND-C/M_Low	CND-G-A14_High	CND-G-A14_Low	CPP_High	CPP_ Low	SWCNT_High	SWCNT_Low
Cyanobacteria (mL)	5.0	5.0	5.0	5.0	5.0	5.0	5.0	5.0	5.0	5.0	5.0
CND (µL)	0.0	4.0	2.0	2.0	1.0	10.0	5.0	0.0	0.0	0.0	0.0
CPP (µL)	0.0	0.0	0.0	0.0	0.0	0.0	0.0	10.0	5.0	0.0	0.0
SWCNT (µL)	0.0	0.0	0.0	0.0	0.0	0.0	0.0	0.0	0.0	56.0	28.0
Total (mL)	5.0	5.004	5.002	5.002	5.001	5.01	5.005	5.01	5.005	5.056	5.028
NP Conc. (ng/µL)	0.0	2149	2149	3943	3943	2030	2030	5000	5000	138	138
Treatment Conc. (ng/mL)	0.0	1718	859	1577	788	4052	2028	9980	4995	1528	768

Notes: CND = carbon nanodot; CND-G = CND synthesized using glucose as the substrate; CND-C/M = CND synthesized using chloroform and methanol as the substrate; CND-G-A14 = fraction A14 derived from a batch of CND-G preparation; CPP = cell-penetrating peptide (γ-Zein-CADY); SWCNT = single-walled carbon nanotube.

## Data Availability

The original contributions presented in the study are included in the article/Appendix A. Further inquiries can be directed to the corresponding author.

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
