# Peer review of "Acute Toxicity of Carbon Nanotubes, Carbon Nanodots, and Cell-Penetrating Peptides to Freshwater Cyanobacteria"

_toxins, 2025, doi:10.3390/toxins17040172_

Round 1
Reviewer 1 Report
Comments and Suggestions for Authors
- The manuscript is well-organized and presents a comprehensive evaluation of acute toxicity of various non-metallic nanoparticles (NMNPs) on nine cyanobacterial strains. The inclusion of multiple endpoints (OD750, OD680, FE450, FE620, QY) strengthens the analysis. However, the authors should briefly mention the ecological relevance of the chosen 48‐hour exposure period. Specifically, how might acute responses compare to longer-term or chronic exposures in natural settings, where factors like nanoparticle persistence, aggregation, and interactions with other biota could alter toxicity profiles
- While the manuscript provides a thorough analysis of NMNP-induced acute toxicity on cyanobacterial growth and physiological endpoints, it does not address whether these nanoparticles influence cyanotoxin production or the release of toxins during the decay of cyanobacterial blooms. Considering that cyanotoxins represent a significant public health and environmental concern, it would be valuable for the authors to discuss, or even investigate for a future work, whether the observed cellular stress or lysis induced by NMNP exposure might trigger the release of intracellular cyanotoxins into the water. Clarification on this aspect—either through a discussion of existing literature or additional experimental data—would greatly enhance the assessment of the potential risks and benefits associated with using NMNPs for bloom mitigation
Author Response
Comment 1: The manuscript is well-organized and presents a comprehensive evaluation of acute toxicity of various non-metallic nanoparticles (NMNPs) on nine cyanobacterial strains. The inclusion of multiple endpoints (OD750, OD680, FE450, FE620, QY) strengthens the analysis. However, the authors should briefly mention the ecological relevance of the chosen 48‐hour exposure period. Specifically, how might acute responses compare to longer-term or chronic exposures in natural settings, where factors like nanoparticle persistence, aggregation, and interactions with other biota could alter toxicity profiles.
Response 1: We accept this suggestion and have added one paragraph to address this comment. Please see the second last paragraph in section 3.4 “Future Research Directions” for details.
Comment 2: While the manuscript provides a thorough analysis of NMNP-induced acute toxicity on cyanobacterial growth and physiological endpoints, it does not address whether these nanoparticles influence cyanotoxin production or the release of toxins during the decay of cyanobacterial blooms. Considering that cyanotoxins represent a significant public health and environmental concern, it would be valuable for the authors to discuss, or even investigate for a future work, whether the observed cellular stress or lysis induced by NMNP exposure might trigger the release of intracellular cyanotoxins into the water. Clarification on this aspect—either through a discussion of existing literature or additional experimental data—would greatly enhance the assessment of the potential risks and benefits associated with using NMNPs for bloom mitigation.
Response 2: We very much appreciate this comment and have added one paragraph to address it. Please see the last paragraph in section 3.4 “Future Research Directions” for details.
Reviewer 2 Report
Comments and Suggestions for Authors
The author compared the cytotoxicity of different materials on various and multiple strains of freshwater cyanobacteria to explore their application in the treatment of cyanobacterial blooms. The treatment of algal blooms and emergency response is a very difficult and urgent task, and there is currently no good governance method. In the research, the author conducted a lot of toxicity testing work, but when applied to on-site algal bloom control, the impact of materials on other phytoplankton, zooplankton, and even fish needs to be considered, which is related to the ecosystem and human health. In addition, I have a few small suggestions for the author's reference:
1. It is recommended to supplement the toxicity testing of other phytoplankton, zooplankton, etc., which is a prerequisite for whether it can be applied on site;
2. In the discussion, it is suggested to supplement the comparison of cost-effectiveness with existing red tide control materials;
3. There are too many images in the article. It is recommended to refine them into no more than 5 images and improve the presentation and clarity of the data.
Author Response
General comments: The author compared the cytotoxicity of different materials on various and multiple strains of freshwater cyanobacteria to explore their application in the treatment of cyanobacterial blooms. The treatment of algal blooms and emergency response is a very difficult and urgent task, and there is currently no good governance method. In the research, the author conducted a lot of toxicity testing work, but when applied to on-site algal bloom control, the impact of materials on other phytoplankton, zooplankton, and even fish needs to be considered, which is related to the ecosystem and human health.
Response: We agree with the reviewer that any treatment methods or technologies for on-site algal bloom control need to assess their impacts on other phytoplankton, zooplankton and even predatory fish should be considered before they are applied to the real-world field. However, this is way beyond the scope of the present study. To address this concern, we did bring this topic up in the Discussion section, present some relevant published studies, briefly discuss the state of art, and identify existing gaps in section 3.4 “Future Research Directions”.
Specific Comment 1: It is recommended to supplement the toxicity testing of other phytoplankton, zooplankton, etc., which is a prerequisite for whether it can be applied on site.
Response 1: Please see our response to the general comments above.
Specific Comment 2: In the discussion, it is suggested to supplement the comparison of cost-effectiveness with existing red tide control materials.
Response 2: We are afraid that this is also beyond the scope of the present study. In particular, it is too early to assess reliably the cost-effectiveness of non-metallic nanoparticle (NMNP)-based remediation technologies for harmful algal blooms control. This is because there are still many uncertainties, non-defined variables and uncharacterized properties or specifications involved in the NMNP technologies. Hence, it is impossible to make any realistic and accurate estimates of cost and efficacy for NMNPs that would allow us to compare with that of existing red tide control materials. Nevertheless, we point out in section 3.4 “Future Research Directions” the needs for investigating the cost-effective of NMNP-based treatment and comparing it with that of existing Cu- and H2O2-based algaecides.
Specific comment 3: There are too many images in the article. It is recommended to refine them into no more than 5 images and improve the presentation and clarity of the data.
Response 3: We respectfully disagree with the reviewer on this point. We have moved as many as 26 non-essential figures and 10 data tables to the Supplementary Materials, including photos of cyanobacterial cultures prior to and post treatments, in-house procedures for preparing and characterizing CNDs and SWCNTs, % change of measurement endpoints, and statistical analysis results. There are a total of 7 figures and 1 table included in the main body of our manuscript. All the 7 figures and 1 table are essential because one won’t be able to comprehend our study and obtained results if any of them is absent. Each of the figures (except Figure 2) comprises of 9 panels, and each panel represents one of the 9 cyanobacterial strains tested in the study. Figure 1 presents the microscopic images of the 9 strains. Figure 2 has five panels, each showing the results of one of the 5 measurement endpoints for three sets of controls. In a logical order, Figures 3 to 7 present the results for 5 measurement endpoints, i.e., OD750, OD680, FE450, FE620, and QY. In terms of presentation clarity, all seven figures are referenced clearly and appropriately in the Results section. Specifically, subsection 2.1 refers to the images in Figures 1 and presents the results in Figure 2; subsection 2.2 focuses on results in Figure 3; subsection 2.3 presents the results in both Figures 4 and 5; subsections 2.4 and 2.5 presents results in Figure 6 and Figure 7, respectively.
Reviewer 3 Report
Comments and Suggestions for Authors
The manuscript investigates the acute cyanotoxicity of non-metallic nanoparticles (NMNPs). The study presents extensive data supporting the effects of NMNPs on nine cyanobacterial species. Below are some minor comments:
- The description of the results is difficult to follow in relation to the figures. The authors should include sub-figure labels and reference them appropriately in the text for better clarity.
- The manuscript repeats sections of the Materials and Methods within the Results section (e.g., lines 93–123). These descriptions should be more concise. Additionally, certain sentences (e.g., lines 149–151) are more suitable for the Discussion section and should be relocated accordingly.
- The supplementary figures illustrating cyanobacterial morphology after NMNP treatment are unclear, making it difficult to follow the explanations in the text. Could the authors specify whether these images were obtained using a light microscope or SEM? Furthermore, figures depicting each type of NMNP are not included. If available, the authors should provide these images.
- What is the scale bar size in Figure 1? Please specify.
Author Response
Comment 1: The description of the results is difficult to follow in relation to the figures. The authors should include sub-figure labels and reference them appropriately in the text for better clarity.
Response 1: We respectfully disagree with the reviewer on this point. All figures (except Figure 2) are organized in a form of 9 panels, each representing one of the 9 cyanobacterial strains tested in the study. Figure 1 presents the microscopic images of the 9 strains. Figure 2 has five panels, each showing the results of one of the 5 measurement endpoints for three sets of controls. In a logical order, Figures 3 to 7 present the results for 5 measurement endpoints, i.e., OD750, OD680, FE450, FE620, and QY. All seven figures are referenced clearly and appropriately in the Results section. Specifically, subsection 2.1 refers to the images in Figures 1 and presents the results in Figure 2; subsection 2.2 focuses on results in Figure 3; subsection 2.3 presents the results in both Figures 4 and 5; subsections 2.4 and 2.5 presents results in Figure 6 and Figure 7, respectively.
Comment 2: The manuscript repeats sections of the Materials and Methods within the Results section (e.g., lines 93–123). These descriptions should be more concise. Additionally, certain sentences (e.g., lines 149–151) are more suitable for the Discussion section and should be relocated accordingly.
Response 2: According to the journal’s requirements, the Material and Method section should appear after the Conclusion section. Therefore, to facilitate the reading flow for the audience, we present in the Results section some of the critical info that will appear later in the Materials and Methods section. Such info (e.g., experimental setup and study design described in lines 93-112) is needed for better comprehending the results presented in the Results section. As for the sentences at the beginning of each of the Results subsections 2.2 through 2.5 (e.g., lines 149-151), they were intended to introduce each of the five measurement endpoints (OD750, OD680, FE450, FE620, and QY) before we present measurement results. This should help the audience to interpret the presented results. Therefore, we don’t think they belong to the Discussion section.
Comment 3: The supplementary figures illustrating cyanobacterial morphology after NMNP treatment are unclear, making it difficult to follow the explanations in the text. Could the authors specify whether these images were obtained using a light microscope or SEM? Furthermore, figures depicting each type of NMNP are not included. If available, the authors should provide these images.
Response 3: The photos in Figures S1 to S10 were taken using a regular digital camera to record the cyanobacterial cultures before (Figure S1) and after treatments (Figures S2 to S10). Our intention was to show the visual appearance of cultures (e.g., turbidity/clearance, color, and aggregation/precipitation). No microscope or SEM was used for imaging. We have drawn molecular structures of CND-G, CND-C/M, and SWCNT, and presented the scheme as Figure S26 in Supplementary Material Part 4. However, Polypeptides like the CPP (γ-Zein-CADY) spontaneously self-assemble into nanoparticles. Therefore, no stable structure could be inferred and presented in this manuscript (see Li T, Lu XM, Zhang MR, Hu K, Li Z. Peptide-based nanomaterials: Self-assembly, properties and applications. Bioact Mater. 2021, 11:268-282. doi: 10.1016/j.bioactmat.2021.09.029) for more info. We have updated the info in subsections 4.2-4.4 of the main body of the manuscript.
Comment 4: What is the scale bar size in Figure 1? Please specify.
Response 4: The scale bar in the original pictures was very vague even after we updated all 9 images in Figure 1. So, we manually redrew the scale bar (10 µm) and inserted it at the bottom right corner in Panel A to be shared by all 9 panels.
Round 2
Reviewer 2 Report
Comments and Suggestions for Authors
The authors had made good revisions and I have no further comments. I suggest the editor consider accepting it.